# Modelling of Proton Exchange Membrane Fuel Cells with Sinusoidal Approach

**DOI:** 10.3390/membranes12111056

**Published:** 2022-10-28

**Authors:** Catalina González-Castaño, Yahya Aalaila, Carlos Restrepo, Javier Revelo-Fuelagán, Diego Hernán Peluffo-Ordóñez

**Affiliations:** 1Centro de Transformación Energética, Facultad de Ingeniería, Universidad Andres Bello, Santiago 7500971, Chile; 2Assistant Investigator Millenium Institute on Green Ammonia as Energy Vector (MIGA), Santiago 7820436, Chile; 3Modeling, Simulation and Data Analysis (MSDA) Research Program, Mohammed VI Polytechnic University, Ben Guerir 47963, Morocco; 4Department of Electromechanics and Energy Conversion, Universidad de Talca, Curicó 3340000, Chile; 5Principal Investigator Millenium Institute on Green Ammonia as Energy Vector (MIGA), Santiago 7820436, Chile; 6Department of Electronics Engineering, Faculty of Engineering, Universidad de Nariño, Pasto 520002, Nariño, Colombia

**Keywords:** Sinusoidal model, proton exchange membrane fuel cell, diffusive model, Evolution Strategy, Gaussian model, voltage-current dynamic response

## Abstract

This paper validates a sinusoidal approach for the proton-exchange membrane fuel cell (PEMFC) model as a supplement to experimental studies. An FC simulation or hardware emulation is necessary for prototype design, testing, and fault diagnosis to reduce the overall cost. For this objective, a sinusoidal model that is capable of accurately estimating the voltage behavior from the operating current value of the DC was developed. The model was tested using experimental data from the Ballard Nexa 1.2 kW fuel cell (FC). This methodology offers a promising approach for static and current-voltage, characteristic of the three regions of operation. A study was carried out to evaluate the effectiveness and superiority of the proposed FC Sinusoidal model compared with the Diffusive Global model and the Evolution Strategy.

## 1. Introduction

Fuel cells have been one of the most researched topics due to their capacity to provide energy for electric vehicles, large-scale energy storage, and power plants [1]. In addition, fuel cells have gained increasing interest as a renewable electric energy source [2]. A fuel cell system is based on the chemical reaction between hydrogen and oxygen or natural air in catalyst cells [3,4]. Among various kinds of fuel cells, the proton exchange membrane fuel cell (PEMFC) has received significant attention from researchers because of its low operating temperature, high power density, high efficiency, fast start-up, and zero pollution emissions [2]. Therefore, the development of an accurate model is necessary to design, simulate, evaluate, and optimize power systems using PEMFC modules [5].

The PEMFC system is a nonlinear, multi-variable system that is difficult to model. Therefore, many models have been developed to simulate and emulate the dynamic nature, as well as to predict PEMFC behavior. Moreover, numerical simulations have been developed to study the effects of the structure and composition on the performance of PEMFCs [6,7,8,9]. In [10], a state-of-the-art review of the models is given. It also outlines, in detail, that the models in [4,11,12,13,14,15,16] consider a linear steady-state response based only on the PEMFC ohmic polarization curve while ignoring the voltage–current dynamic response. Therefore, these models are not viable for analyzing the complete polarization curve of an FC. On the other hand, most models use an analytical approach based on the physical system of the FC. These approaches need many variables to evaluate the model. However, due to their implementation complexity requirements, the use of multiple variables for evaluation increases the development costs and the requirements of a high-processing device.

A novel FC model based on a Sinusoidal approach is presented in this paper. This model is a multi-parametric method solved by approximation to a first-order Taylor polynomial expansion. The aim was to construct a mathematical model based on a weighted combination of sine functions to reflect the intricacies of FC system behavior under the considered working conditions. A multi-parametric Gaussian model for FC was introduced in [10]. One advantage of the sinusoidal model over its Gaussian counterpart is that the former, in theory, requires fewer function terms. One sine function can model two peaks, whereas the Gaussian model needs twice as many terms. Furthermore, Sinusoidal-based models can fit both periodic and repetitive behaviors, which means that fewer parameters are needed for estimation in comparison with Gaussian peaks [10].

The model includes the fuel cell’s steady state and transient responses. Moreover, the model only needs the operating current of the fuel cell to predict the output voltage behavior; this is true for large-signal, step-type variations at any point of the whole operation range of the current. Thus, the model is suitable for implemention in a low-cost Digital Signal Controller (DSC). In this work, the proposed Sinusoidal model was built, optimized, and tested with data obtained from a Nexa power module from Ballard Power Systems. The experimental results show that the cell voltage was predicted with a fast dynamic response and good steady-state accuracy.

The following are the key features of this work:A Sinusoidal FC model was developed to identify the fuel cell behavior for a profile of the operating current of a fuel cell for steady-state and dynamic responses.Experimental data from a commercial Nexa fuel Cell Power module were used to train and estimate model parameters that best fit the different profiles for testing.The results were compared using analytical and numerical techniques under the same data acquisition parameters to ensure a fair comparison between the models.The obtained results prove the effectiveness of the proposed FC model compared with the Evolution strategy [17], the diffusive model [18], and the Gaussian model [10].

This paper is structured as follows: Section 2 outlines the multi-parametric Sinusoidal model considered in this work. Next, Section 3 gathers the experimental results and presents the discussion. Finally, the conclusions are presented in Section 4.

## 2. Sinusoidal Model

Curve fitting is essentially used in any task in which the construction of a mathematical function to fit experimental data is required. In particular, parametric curve fitting [19] is used in various real-life problems, such as modeling the growth of a population, the spread of infectious diseases, and the relationship between crop growth and yield in the contexts of ecology, epidemiology, and agriculture, respectively. It assumes a function form f(.,θ) to best fit the data by finding the optimal value of the parameter θ.

In this work, we used the sinusoidal model, which is described by g(x,θ) in Equation (Equation 1), where $\theta =\left(a,b,c\right)\in R^{3}$ is the parameter to estimate. The terms *a*, *b*, and *c* stand for the amplitude, frequency, and the phase constant, respectively.
(1)$$\begin{array}{c} g\left(x,\theta \right)=a\sin\left(bx+c\right) \end{array}$$

In Figure 1, we convey an idea about the parameters’ effects on g(x,θ) by varying *a*, *b*, and *c*.

Based on Equation (Equation 1), the generalized mathematical expression of the sinusoidal model is based on the following function:(2)$$\begin{array}{cc} f_{\sin}\left(x,\theta \right) & =\sum_{j=1}^{N}g_{j}\left(x,\theta_{j}\right)\\ \; & =\sum_{j=1}^{N}a_{j}\sin\left(b_{j}x+c_{j}\right) \end{array}$$
where $\theta ={\left(\theta_{j}\right)}_{1\leq j\leq N}\in R^{3N}$, with $\theta_{j}=\left(a_{j},b_{j},c_{j}\right)$ which holds the amplitude, frequency, and the phase constant, respectively, for each sine function term. *N* is the number of terms involved in the series. Using *N* sine terms in Equation (Equation 2) serves to depict the different types of peaks in any given dataset, especially if the data are approximating periodic or recurrent behaviors.

Figure 2 illustrates two specific examples of $f_{\sin}\left(x,\theta \right)$’s resulting curve, where the estimated parameters are slightly different and depict different peaks.

The purpose of this work was to estimate the aforementioned parameter θ to determine how to best fit a given set of data points $\left(x_{i},y_{i}\right)$, 1≤i≤m, where *m* is the number of samples. To that end, the unconstrained optimization problem is stated as follows:(3)$$\begin{array}{c} \min_{\theta \in R^{3N}}\;\sum_{i=1}^{m}r_{i}{\left(\theta \right)}^{2}=\min_{\theta \in R^{3N}}\;\sum_{i=1}^{m}{\left(y_{i}-f_{\sin}\left(x_{i},\theta \right)\right)}^{2}, \end{array}$$
where $r_{i}$ is the residual between $y_{i}$ and its estimate $f_{\sin}\left(x_{i},\theta \right)$.

As $f_{\sin}\left(x_{i},\theta \right)$ is defined in Equation (Equation 2), the above optimization problem is a non-linear least squares (LS) formulation. For that reason, a closed form of the solution is generally not feasible and other refined iterative approaches are needed.

In this work, the widely-used Gauss–Newton approach [20], where the model function is linearized by approximation to a first-order Taylor polynomial expansion, was considered. Generally, the core of the iterative approach relies on updating the parameter $\theta^{\left(k\right)}$, for a given iteration *k* in the following form:(4)$$\begin{array}{c} \theta^{\left(k+1\right)}=\theta^{\left(k\right)}+d_{k},\;\mathrm{with}\;d_{k}=-{\left[J{\left(\theta^{\left(k\right)}\right)}^{⊤}J\left(\theta^{\left(k\right)}\right)\right]}^{-1}J{\left(\theta^{\left(k\right)}\right)}^{⊤}r\left(\theta^{\left(k\right)}\right), \end{array}$$

The term $J(\theta^{\left(k\right)})$ stands for the Jacobi matrix which, in various cases, suffers from ill-conditioning problems [21], as the inverse is involved in the updating formulation. Thus, for reliability, robustness, and convergence properties, the classical trust region-based approach of the Levenberg–Marquardt (LM) method [22] for nonlinear LS optimization problem was implemented, instead of a conventional line search approach.

The LM formulation relies on the parameters being updated based on a trust region; that is, given an initial neighborhood $B(\theta_{0},\lambda_{0})$ of an initial guess $\theta_{0}$, an approximate model is constructed, and it is solved in a constrained manner in $B(\theta_{0},\lambda_{0})$, where *B* is a ball in some norm with the radius $\lambda_{0}$. This is updated repeatedly until convergence [22]. Equation (Equation 4), in this context, becomes
(5)$$\begin{array}{c} \theta^{\left(k+1\right)}=\theta^{\left(k\right)}-{\left[J{\left(\theta^{\left(k\right)}\right)}^{⊤}J\left(\theta^{\left(k\right)}\right)+\lambda_{k}I\right]}^{-1}J{\left(\theta^{\left(k\right)}\right)}^{⊤}r\left(\theta^{\left(k\right)}\right), \end{array}$$
where, I is a 3N×3N identity matrix $\theta^{\left(k\right)}=\left(\theta_{j}^{\left(k\right)}\right)$ with $\theta_{j}^{\left(k\right)}=\left(a_{j}^{\left(k\right)},b_{j}^{\left(k\right)},c_{j}^{\left(k\right)}\right)$. Additionally, $J(\theta^{\left(k\right)})$ is the Jacobian of $r(\theta^{\left(k\right)})$, which is calculated as follows:(6)$$\begin{array}{c} J\left(\theta \right)=\left[\begin{array}{cccc} \frac{\partial r_{1}\left(\theta \right)}{\partial \theta_{1}} & \frac{\partial r_{1}\left(\theta \right)}{\partial \theta_{2}} & \cdots & \frac{\partial r_{1}\left(\theta \right)}{\partial \theta_{n}}\\ \frac{\partial r_{2}\left(\theta \right)}{\partial \theta_{1}} & \frac{\partial r_{2}\left(\theta \right)}{\partial \theta_{2}} & \cdots & \frac{\partial r_{2}\left(\theta \right)}{\partial \theta_{n}}\\ \vdots & \vdots & \ddots & \vdots \\ \frac{\partial r_{m}\left(\theta \right)}{\partial \theta_{1}} & \frac{\partial r_{m}\left(\theta \right)}{\partial \theta_{2}} & \cdots & \frac{\partial r_{m}\left(\theta \right)}{\partial \theta_{n}} \end{array}\right], \end{array}$$
with
$$\frac{\partial r_{i}\left(\theta \right)}{\partial \theta_{j}}=\left[\sin\left(b_{j}x_{i}+c_{j}\right),\;a_{j}x_{i}\cos\left(b_{j}x_{i}+c_{j}\right),\;a_{j}\cos\left(b_{j}x_{i}+c_{j}\right)\right].$$

Algorithm 1 summarizes the steps taken to reach an adequate estimate of the parameters of the Sinusoidal model.
**Algorithm 1:**Unconstrained nonlinear optimization procedure**Input:** Measured dataset ${\left\{(x_{i},y_{i})\right\}}_{i=1}^{m}$
   1: Use the mathematical model defined by Equation (Equation 2)
   2:Calculate the residual vector’s entries $r_{i}\left(\theta \right)=y_{i}-f_{\sin}\left(x_{i},\theta \right)$
   3:Determine the Jacobian matrix J(θ)
   4:Use a Non-linear Least Squares algorithm to estimate the optimal parameters, as described in Equation (Equation 5)
**Output:** The vector parameter θ


Finally, the mathematical statements presented above are expressions in terms of the generic independent variable *x*, and the correspondence of variables for FC purposes is mentioned below:*x*: electric current *I*,$f_{\sin}\left(x\right)$: voltage $v=f_{\sin}\left(I\right)$.

## 3. Experimental Results

The system used to obtain the experimental data and to produce the modes was a Nexa fuel cell. This cell generates unregulated DC power from hydrogen and air. In addition, the Nexa power module has a graphical user interface to show the operational status and performance [23]. Figure 3 shows the experimental Nexa PEMFC data acquisition configuration used for the training and validation of the Sinusoidal model.

The LeCroy WaveSurfer 64Xs-A oscilloscope has fast acquisition, long capture time, and data saving properties on its onboard hard drive. An oscilloscope directly acquires and stores the data corresponding to the fuel cell current and voltage signals. Thus, the maximum sampling limitation of the Nexa software is avoided, achieving sampling periods of up to 20 μs. Furthermore, a virtual instrument was developed using LabVIEW, which generates the current profiles through the DC electronic load control using its GPIB communication port as a constant current load.

### 3.1. Training Model Used in the Fuel Cell System

The sinusoidal model was used to estimate the voltage in a different current region of the fuel cell. Specifically, the Nexa fuel cell case that has current training data between 0 A and 45 A. FC characterization is given by the curve shown in Figure 4, while the dynamic V-I characteristic of the PEMCF is given using the real FC voltage and estimated FC voltage from the proposed sinusoidal model with the Expression (Equation 7). The FC voltage was predicted for each experimental current data point shown in Figure 5. The Sinusoidal model responded accurately with real data from the fuel cell. A FC load current profile was generated, as shown in Figure 5, to reproduce the different operating points and transients to train the model in all operating current FC subdomains.

It can be observed that the load profile is provided in A (current) instead of A/cm${}^{2}$ (current density of active area). It is well-known that the current density allows an easy comparison between different FC systems. However, information about the Nexa active area is not provided by the manufacturer. Meanwhile, the literature reports different values such as 100 cm${}^{2}$ in [24] and 110 cm${}^{2}$ in [25], among others.

As described in (Equation 2), an important parameter to determine when using the Sinusoidal model is *N*, which stands for the number of terms involved in the sum. Practically speaking, *N* ranges from 1 to 8, as any additional terms may lead to either overfitting problems or a lack of substantial improvement in the model’s performance. In this work, the number of terms chosen was N=6, as the corresponding model performed best during the validation phase. To that end, the resulting Sinusoidal model, outlined in (Equation 7), includes six sine functions. This means that 18 parameters were used for the estimation using the unconstrained nonlinear optimization procedure detailed in Algorithm 1.
(7)$$\begin{array}{cc} f_{\sin}\left(I\right)= & \;\;57.54\cdot \sin\left(0.05466\cdot I+0.7158\right)\\ \; & +27.32\cdot \sin\left(0.09958\cdot I+2.95\right)\\ \; & +8.865\cdot \sin\left(0.2489\cdot I+2.901\right)\\ \; & +6.232\cdot \sin\left(0.2798\cdot I+5.395\right)\\ \; & +0.5072\cdot \sin\left(0.3939\cdot I+6.231\right)\\ \; & +0.3487\cdot \sin\left(0.8202\cdot I+0.3831\right), \end{array}$$
where *I* is the current data input from the FC. In this context, Equation (Equation 7) provides the estimation of the voltage given the current I. It can be observed that, as the current drawn from the FC increased, the FC voltage decreased. Additionally, the simulated FC voltage closely followed the experimental FC voltage. A high deviation between the experimental voltage data and the values estimated by the model occurred at 3410.74 s, corresponding to a current input of 44.45 A. At this point, the voltage difference was 3 V, corresponding to a relative error of 13.37%. The modeling results were quantified using the root mean square error (RMSE). The obtained results are considered satisfactory, given the fact that, for the proposed model, the RMSE was 0.44 V.

### 3.2. Validating Model in the Fuel Cell System

To validate the proposed approach, the model entered a testing phase, in which different experimental data were used. The current profile was significantly challenging as opposed to that used in training. As illustrated in Figure 6, this was due to the higher step changes in terms of the current magnitude. The proposed model was compared with the diffusive approach from [18] and the Gaussian model proposed in [10]. The maximum deviation for the models occurred at 518.14 s at 38.42 A. The difference between the sinusoidal model’s estimation and the experimental values was 2.5 V with a relative error of 9%. For the Gaussian model voltage value and the experimental one, the difference was 2.86 V, with a relative error of 11.97%. For the diffusive model, the voltage difference was 4.87 V with a relative error of 24.7%. Therefore, the Sinusoidal model fits the experimental data for validation better than both the Gaussian and diffusive models with an RMSE of 0.53 V. The Gaussian and diffusive models scored RMSE values of 0.65 V AND 1.05 V, respectively. Figure 7 shows an analytical spider chart of the FC models regarding different statistical indicators.

The statistical indicator results presented in Figure 7 demonstrate that the proposed Sinusoidal model produced a low error rate and a high R-square value compared with the Diffusive global model and the Gaussian model. The Gaussian model produced a lower value for the SD criteria only.

### 3.3. Comparison of the Sinusoidal Model with the Parameter Identification by Means of the Evolution Strategy, Diffusive Global Model, and Gaussian Model

The model based on the parameter identification of an equivalent circuit-based proton-exchange membrane fuel cell model was introduced in [17] using an ES. Training and validation data were sampled within a period of 200 ms in [17], which is 10 times greater than the sampling period used in the profile outlined in Figure 5. Therefore, the models were, again, validated and trained. The current experimental data profile used for training the Sinusoidal model can be observed in Figure 8a, and the Diffusive Global model is presented in [18]. The training model results are shown in Figure 8b. This figure illustrates the experimental response of the fuel cell to the load current profile shown in Figure 8a. The RMSE of the Diffusive Global model was found to be 0.3648 V. For the parameters adjusted by ES, the RMSE was 0.7961 V, for the Gaussian approach, it was 0.3878 V, and for the Sinusoidal model, it was 0.3574 V. Therefore, the diffusive approach, the Gaussian model, and the sinusoidal model produce similar predictions of the output voltage.

Finally, Figure 9 shows the validation of the approaches, where the RMSE of the diffusive global model and the parameters adjusted by ES were 2.3273 V and 2.3116 V, respectively. The RMSE for the Gaussian approach was 0.47 V, and for the Sinusoidal model, it was 0.28 V. The statistical results of the proposed Sinusoidal model are detailed in Figure 10. These results were compared with those from the Diffusive global model, Gaussian model, and the ES approach. The Sinusoidal model was found to have the lowest error value. This result demonstrates that the proposed Sinusoidal model is capable of accurately estimating the voltage of the FC, even under the most challenging circumstances, as it accurately represents the static and dynamic current–voltage relationship in a PEMFC.

## 4. Conclusions

In this paper, a PEMFC model with the Sinusoidal approach was introduced. The model was designed to estimate the FC voltage for the steady-state and dynamic responses. The results from the proposed model show similar behaviors to the results from the experimental data with the Ballard Nexa 1.2 kW FC. Different training and validation profiles were developed to compare the proposed model with the Diffusive global model, Gaussian model, and the Evolution strategy based on numerical and analytical techniques. The comparison results show that the Sinusoidal model produced a superior performance with good steady-state accuracy and easy implementation with previous adjustment of its parameters.

## Figures and Tables

**Figure 1 membranes-12-01056-f001:**
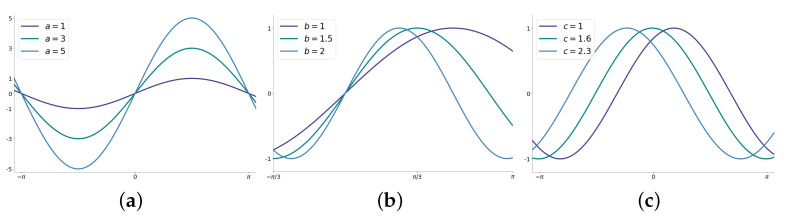
Effects of the terms *a*, *b* and *c* in Equation (Equation 1): (**a**) variations for the parameter *a*, (**b**) variations for the parameter *b* and (**c**) variations for the parameter *c*.

**Figure 2 membranes-12-01056-f002:**
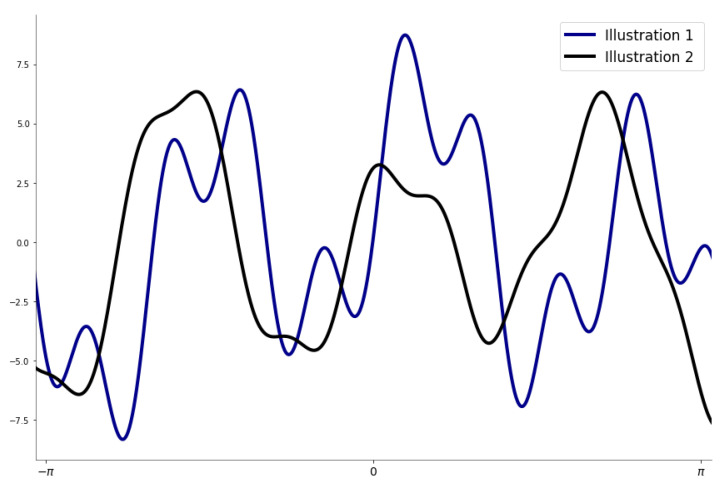
Resulting curve of the three-terms sum of the sine model with slightly different parameters.

**Figure 3 membranes-12-01056-f003:**
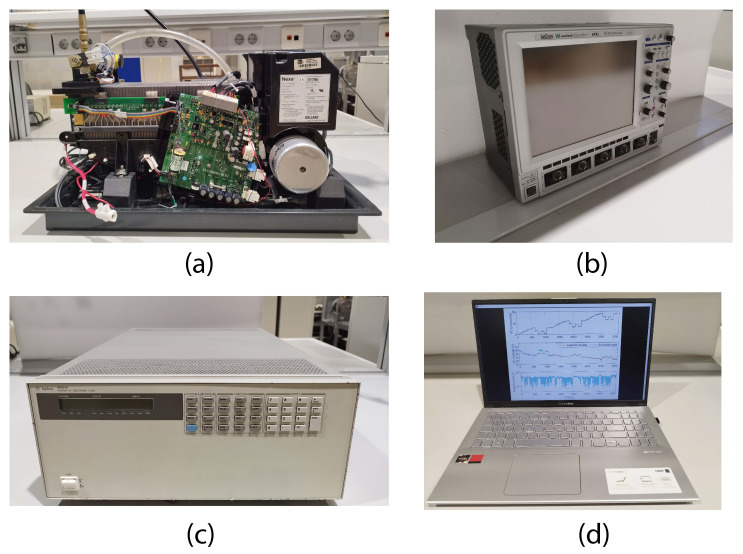
Experimental data acquisition configuration used for the Sinusoidal model training and validation: (**a**) Nexa power module from Ballard Power Systems, (**b**) LeCroy WaveSurfer 64Xs-A oscilloscope that saves trace data to an internal memory location, (**c**) Agilent 6050A Electronic load, (**d**) Laptop computer + LabVIEW program used to control the electronic load using the National Instruments GPIB-USB-HS+ software program to monitor the Nexa Fuel Cell.

**Figure 4 membranes-12-01056-f004:**
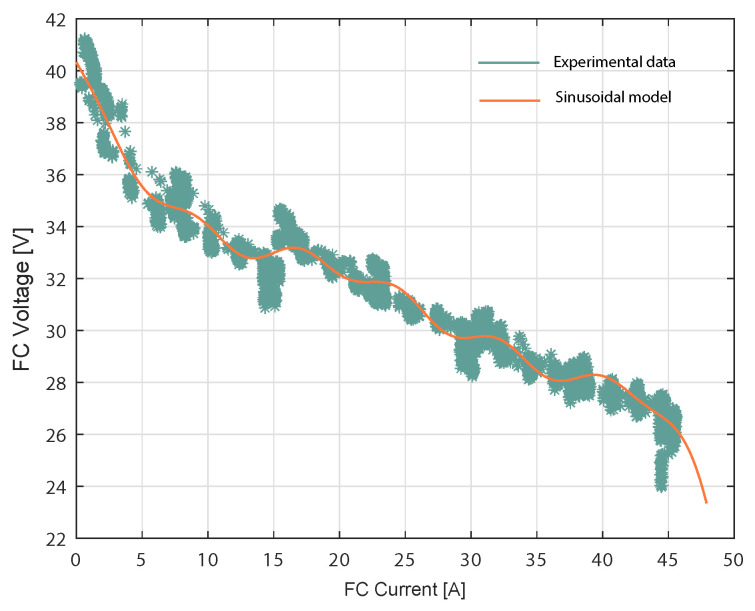
V-I characteristics of the FC and Sinusoidal model.

**Figure 5 membranes-12-01056-f005:**
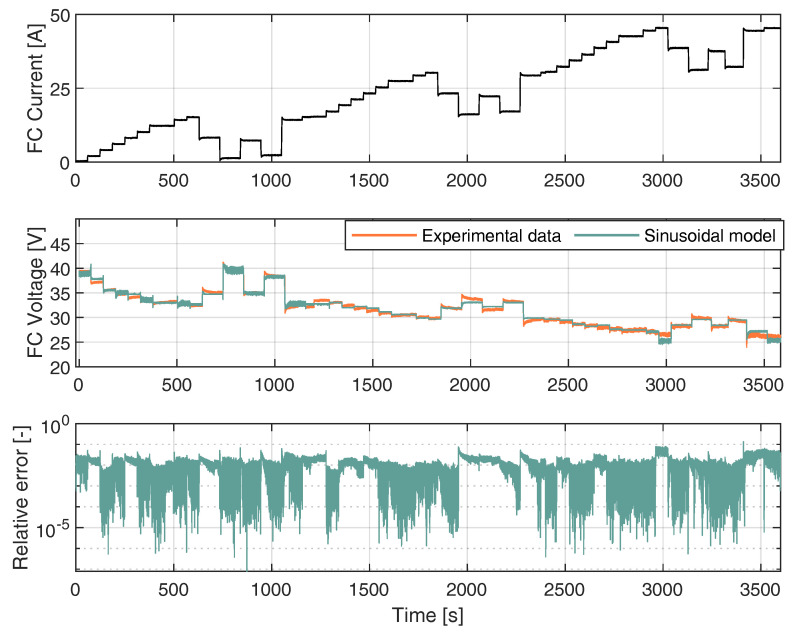
Training the FC Sinusoidal model.

**Figure 6 membranes-12-01056-f006:**
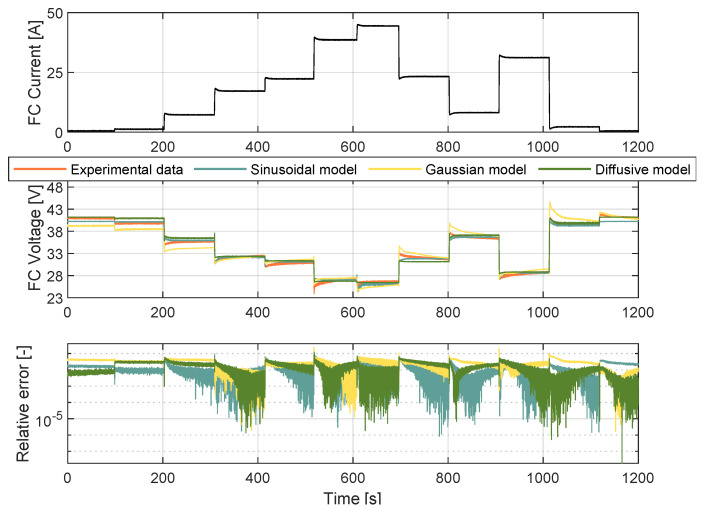
Validating the FC Sinusoidal model.

**Figure 7 membranes-12-01056-f007:**
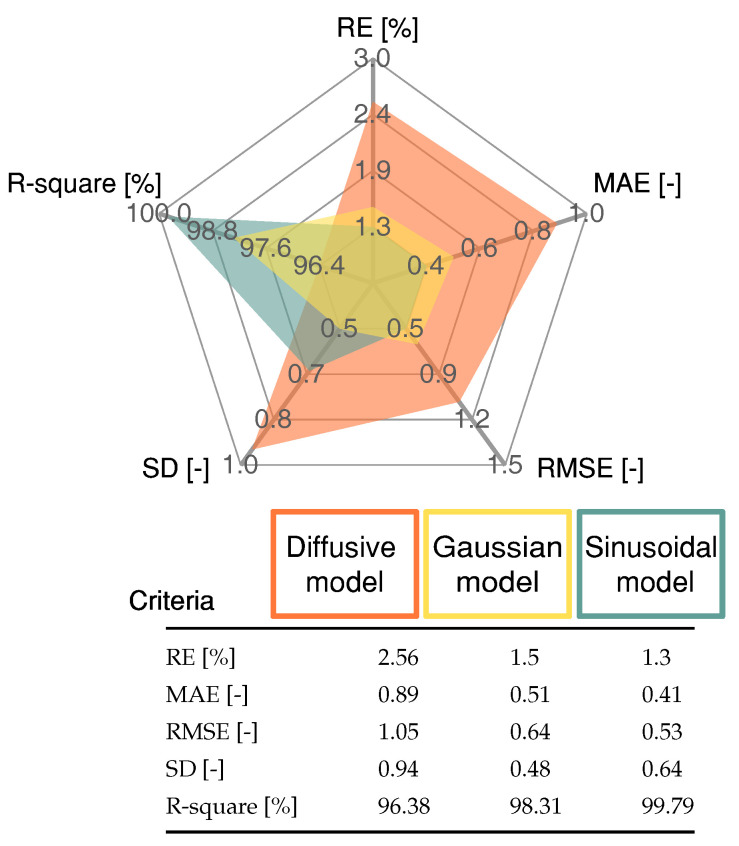
Statistical results of the proposed Sinusoidal model, Gaussian model and the Diffusive global model for the profile shown in Figure 6. To this end, four error measures are reported namely, Relative error (ER), Mean absolute error (MAE), Root mean square error (RMSE) and Standard deviation (SD).

**Figure 8 membranes-12-01056-f008:**
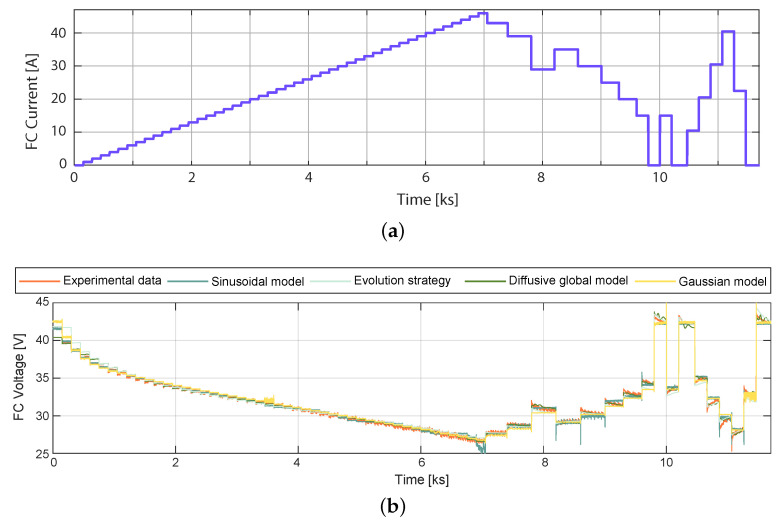
Experimental Nexa FC data used for training: (**a**) current load profile, (**b**) output voltage simulated with parameters estimated by means of the ES, the diffusive global model, the Gaussian model, and the Sinusoidal model.

**Figure 9 membranes-12-01056-f009:**
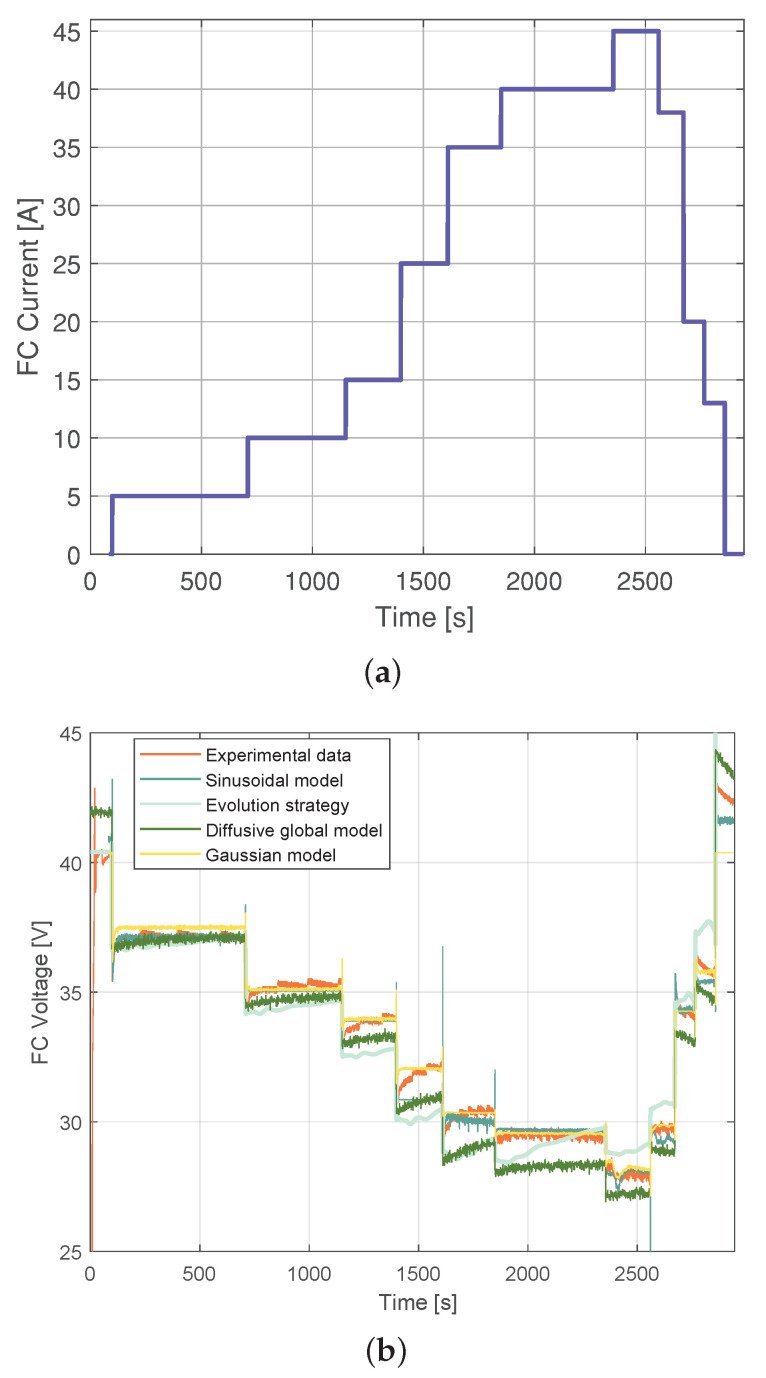
Experimental Nexa FC data used for validation: (**a**) current load profile and (**b**) output voltage simulated with parameters estimated by means of ES, the diffusive global model, the Gaussian model, and the Sinusoidal model.

**Figure 10 membranes-12-01056-f010:**
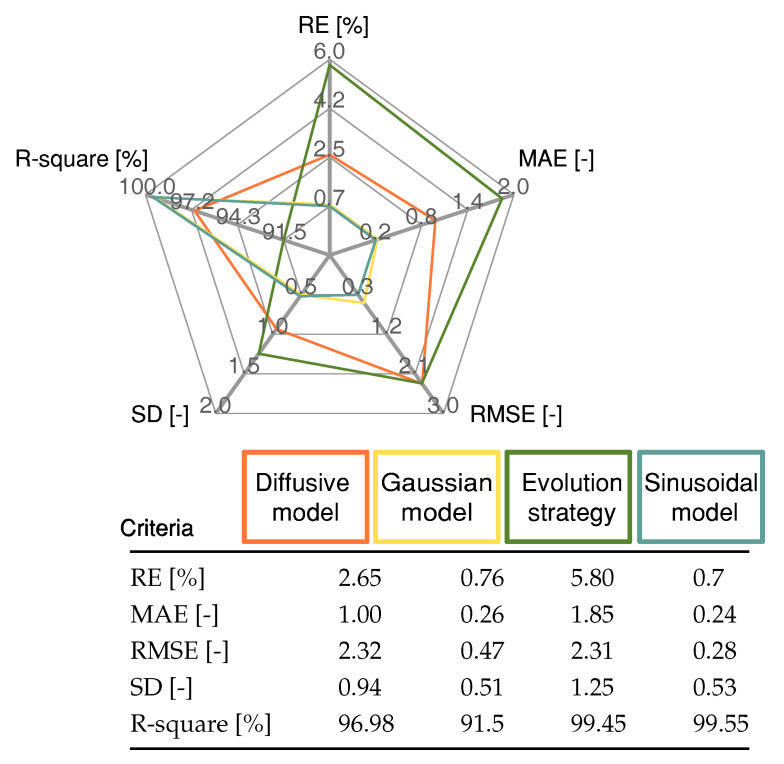
Statistical results for the proposed Gaussian model, Diffusive global model, and ES approach for the profile shown in Figure 9.

## Data Availability

Not applicable.

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
