# Peer review of "Modelling of Proton Exchange Membrane Fuel Cells with Sinusoidal Approach"

_membranes, 2022, doi:10.3390/membranes12111056_

Round 1

Reviewer 1 Report

The work presents a model for proton exchange membrane fuel cells that considers the dynamic behaviour of the fuel cell, which is a very important aspect to understand the experimental response of a fuel cell. The work is similar to one published for the authors, reference [4], but the V-I  fuel cell response  is expressed in this case as a sum of sinusoidal functions, instead of Gaussian functions. The work is interesting and suitable of publication in Membranes, but it should be improved.

-The selection of the value of N is not clear in the ext.

-The method to obtain curves versus time from V-I characteristic should be better explained in the text. It is an important aspect that is not enough clear in the text.

-Figure 4 must be placed after Figure 3 in the text.

-Define all  abbreviations in Figs. 7 and 10

-In pag.8, line 169, “ the trained mode results are showed in Fig.8(b)“,  instead   of Fig. 8 ?

Author Response

-The selection of the value of N is not clear in the text.
**Response: As recommended rightfully by the reviewer, a paragraph was added in lines 133-140 outlining the reasoning
behind the choice of N.
-The method to obtain curves versus time from V-I characteristic should be better explained in the text. It is an important
aspect that is not enough clear in the text.
**Response: Thank you very much for the comment. We modified the description text about the method to obtain the V-I
characteristic shown in Figure 4.
-Figure 4 must be placed after Figure 3 in the text.
**Response: As rightfully suggested by the reviewer, Figure 4 is placed after Figure 5.
-Define all abbreviations in Figs. 7 and 10.
**Response: As suggested by the reviewer, the four abbreviations mentioned in Fig. 7 and 10 are detailed in the respective
captions. In addition, a list of abbreviations used in the manuscript is detailed in page 10, line 213.
-In pag.8, line 169, “the trained mode results are showed in Fig.8(b)“, instead of Fig. 8 ?
**Response: Indeed, as suggested by the reviewer, we rectified the hyper-reference to be specific to sub-figure (b) of Fig.8
(Fig.8(b)) .      

Reviewer 2 Report

The work has been carried out with care. Overall, the work appears to be of good quality and the manuscript is well presented

Author Response

-The work has been carried out with care. Overall, the work appears to be of good quality and the manuscript is well
presented.
**Response: Thank you very much for your comments, we appreciate the positive feedback by the reviewer.      

Reviewer 3 Report

The manuscript evaluates “Modelling of proton exchange membrane fuel cells with sinusoidal approach.

Catalina and co-authors demonstrate interesting work.

Article required revision before considering publication in the Membranes.

Limitation of the work, novelty and contributions should be highlighted more.

In the introduction section still a lack of literature is missing & need to rewritten about metal based various Modelling simulation applications, authors should read below relevant reference and consider for citation.

Membranes 202212(10), 1001. Journal of Applied Polymer Science 108 (6), 3572-3576, 2008.

The presented Sinusoidal results agree with experimental data with 1.2kW FC.

Authors should also read recent articles of  Membranes journal and update the same in the revised version.

Overall the article is fairly well written, after addressing the above comments the article may be considered for publication.

English and grammatical errors should be rectified during the revision of the paper.

Author Response

-Limitation of the work, novelty and contributions should be highlighted more.
**Response: Thank you for you insightful comment. In the introduction - page 2 lines 52- 61, four points where mentioned
in which the authors tried to highlight the major contributions. In this light and as recommended by the reviewer, throughout
the manuscript, modifications and rephrasing were made -when applicable- in the attempt to highlight the novelty and
contributions intended by this work.
-In the introduction section still a lack of literature is missing need to rewritten about metal based various Modelling
simulation applications, authors should read below relevant reference and consider for citation.
**Response: We would like to thank the reviewer for his constructive and valuable comments. Following your suggestions,
the authors want to thank the reviewer for the proposed references that have been very useful in improving the quality of the
article. The introduction was improved by adding the references [R1] and [R2] to present PEMFC models for simulation
applications.
• -[R1] Zhan, Z.; Song, H.; Yang, X.; Jiang, P.; Chen, R.; Harandi, H.B.; Zhang, H.; Pan, M. Microstructure
Reconstruction and Multiphysics Dynamic Distribution Simulation of the Catalyst Layer in PEMFC. Membranes 2022,
12, 1001.
• -[R2] Jawalkar, S.; Nataraj, S.; Raghu, A.; Aminabhavi, T. Molecular dynamics simulations on the blends of poly (vinyl
pyrrolidone) and poly (bisphenol-A-ether sulfone). Journal of Applied Polymer Science 2008, 108, 3572–3576
-Authors should also read recent articles of Membranes journal and update the same in the revised version.
**Response: We appreciate the reviewer’s comments and suggestions. We added the following references from Membranes
journal about different PEMFC models and its applications.
• -[R3] Li, Y.; Ma, Z.; Zheng, M.; Li, D.; Lu, Z.; Xu, B. Performance analysis and optimization of a high-temperature
PEMFC vehicle based on particle swarm optimization algorithm. Membranes 2021, 11, 691
• -[R4] Dickinson, E.J.; Smith, G. Modelling the proton-conductive membrane in practical polymer electrolyte membrane
fuel cell (PEMFC) simulation: A review. Membranes 2020, 10, 310
• -[R5] Wang, W.; Qu, Z.; Wang, X.; Zhang, J. A molecular model of PEMFC catalyst layer: simulation on reactant
transport and thermal conduction. Membranes 2021, 11, 148.
-English and grammatical errors should be rectified during the revision of the paper.
**Response: As rightfully suggested by the reviewer, the entirety of the manuscript was scanned for grammatical errors,
spelling check and style refinement.      

Round 2

Reviewer 1 Report

The paper has been modified according to the comments.